

# Urbanization, environmental stabilization and temporal persistence of bird species: a view from Latin America

Lucas Matías Leveau

Departamento de Ecología, Genética y Evolución, Facultad de Ciencias Exactas y Naturales, Universidad de Buenos Aires—IEGEBA (CONICET-UBA), Ciudad Universitaria, Buenos Aires, Argentina

## ABSTRACT

**Background**. A scarcely studied consequence of urbanization is the effect of temporal stabilization of the environment on bird communities. This alteration is thought to dampen environmental variations between day and night, seasons and years, promoting a temporal persistence of bird composition in urban areas. The aim of this study was to review current evidence of temporal stabilization of biotic and abiotic factors in urban environments and the potential effects of such stabilization on temporal variation of bird species presence at different temporal scales.

**Methods**. I selected the literature by searching published articles and book chapters using Scopus and Google scholar. I only included articles that compared the temporal variation of bird composition or resources between different levels of urbanization.

**Results**. In general, there is evidence of temporal stabilization of abiotic and biotic factors at the three time scales considered. At the diurnal scale, the main factor considered was artificial light in the context of light pollution. At the seasonal and interannual scales, several case studies found a smaller temporal variation of primary productivity in urban than in natural and rural areas. Bird species composition showed more stabilization in urban environments at the three temporal scales: (1) several case studies reported bird activity at night, associated with artificial light; (2) studies in urban parks and along urbanization gradients showed smaller seasonal variation of bird composition in the more urbanized areas; and (3) in general, case studies along urbanization gradients showed smaller interannual variation of bird composition in the more urbanized areas, although some studies showed no relationships or opposite trends than expected.

**Discussion**. The published evidence suggests that urban areas dampen the natural cycles at several temporal scales. The stabilization of biotic and abiotic factors, such as light, temperature, food and habitat structure, is desynchronized from natural diurnal, seasonal and interannual cycles. However, there is a dearth of long-term comparisons of bird composition and studies that simultaneously analyze the relationship between resources and bird composition stabilization at the seasonal and interannual scales. More research is needed in the Southern hemisphere, where there is a lack of studies dealing with the seasonal and interannual variations of primary productivity along urbanization gradients and nocturnal activity of bird species. A future research agenda should include differentiation of spatial and temporal homogenization of avifaunas.

Corresponding author
Lucas Matías Leveau,
leveau@ege.fcen.uba.ar,
lucasleveau@yahoo.com.ar

## INTRODUCTION

Urbanization has many impacts on natural and semi-natural environments; urban expansion promotes fragmentation of ecosystems, perforation through the creation of different urban areas, alteration of biogeochemical cycles, the occurrence of the urban heat island phenomenon and pollution (*Miller et al., 2001*; *Grimm et al., 2008*; *Shanahan et al., 2014*). These environmental changes have impacts on the spatial dynamics of species, producing significant declines in species diversity in highly urbanized areas and significant changes in bird composition (*Faeth, Bang & Saari, 2011*; *Suarez-Rubio, Leimgruber & Renner, 2011*). The uniform structure of urban areas across the world promotes the invasion of a reduced number of cosmopolitan species, leading to biotic homogenization (*McKinney, 2006*). A scarcely explored consequence of urbanization is the temporal stabilization of biotic and abiotic factors, which may induce a decline of the temporal beta diversity of biological communities.

Birds are one of the most widely studied taxa in urban environments because they are easily observable and respond to environmental changes (*Lepczyk et al., 2017*). Birds can be used as indicator species of habitats that support other species and, therefore, contribute to the understanding of the impact of urbanization on biodiversity (*Gil & Brumm, 2014*; *Lepczyk et al., 2017*). In addition, birds are associated with ecosystem services and disservices (*Sekercioglu, 2006*; *Lyytimäki et al., 2008*; *Wenny et al., 2011*; *Belaire et al., 2015*).

In natural areas, biotic factors such as predator presence or food availability, and abiotic factors, such as the variation of light throughout the day or seasonal changes in rain or temperature, induce temporal changes in species composition, favoring the coexistence of species through temporal segregation and an increase in beta diversity (*Schoener, 1974*; *Herrera, 1978*; *Kronfeld-Schor & Dayan, 2003*; *Uchida & Ushimaru, 2015*). For example, light variation during the day allows the coexistence of diurnal and nocturnal raptors in a given area, reducing their agonistic interactions (*Jaksić, 1982*). Seasonal fluctuations in the amount of resources favor the coexistence of resident and migrant species in temperate regions (*Hurlbert & Haskell, 2003*; *Dalby et al., 2014*).

Recently, several authors noted that urban areas may promote a significant loss of temporal heterogeneity of biological diversity (*Suhonen et al., 2009*; *Leveau & Leveau, 2012*; *La Sorte, Tingley & Hurlbert, 2014*; *Leveau, Isla & Bellocq, 2015*; *Uchida, Fujimoto & Ushimaru, 2018*), presumably as a consequence of the stabilization of biotic and abiotic factors. Dampening of temporal variation in resources may influence bird community composition by favouring the temporal persistence of bird species capable of exploiting such resources, the so-called urban exploiters and adapters (see *Blair, 1996*). Furthermore, stabilization of habitats and resources may promote the local extinction of those species adapted to temporal changes of resources and natural disturbances (*Gliwicz, Goszczyński & Luniak, 1994*; *Luniak, 2004*; *Shochat et al., 2006*; *Duckworth, 2014*; *Parris, 2016*; *Pickens*

*et al., 2017*). For example, the Bachman's sparrow (*Peucaea aestivalis*) is related to fire disturbance, and a decline in its numbers is predicted due to projected urban growth (*Pickens et al., 2017*). The ultimate consequences of these changes would be the local dominance and regional expansion of species benefited by the stabilization of resources in urban environments (*Shochat et al., 2006*; *Duckworth, 2014*; *Parris, 2016*).

Stabilization of urban environments needs to be addressed at several temporal scales because urbanization may promote the dampening of environmental conditions between day and night, seasons and years. For example, the profound alteration produced by artificial light in urban environments at night is associated with nocturnal activity of birds (*Rejt, 2004*; *La, 2012*). The reduced annual variation of food resources in urban areas has a negative effect on the presence of migratory species and favors the permanence of resident bird species (*Leveau, Isla & Bellocq, 2015*; *Leveau & Leveau, 2016*). Moreover, the management of interannual natural disturbances, such as flood or fire, in urban areas may lead to the extinction of bird species associated with the changes in the landscape induced by such disturbances (*Pickens et al., 2017*). Temporal stabilization of the environment in urban areas may induce a temporal homogenization of bird communities; under this scenario, humans perceive a similar bird composition at any time and can be disconnected from the natural rhythms of nature (*Leveau, Isla & Bellocq, 2015*; *Leveau & Leveau, 2016*).

Most reviews about urban bird ecology were conducted in developed countries (*Chace & Walsh, 2006*; *Shanahan et al., 2014*; *Reynolds et al., 2017*). Cities in developing regions, such as in Latin America, have socioeconomic and morphological contrasts with cities in developed countries. For example, Latin American countries have lower per capita income, higher socioeconomic inequality, and more compact and dense cities (*Huang, Lu & Sellers, 2007*) than developed countries. These socioeconomic and morphological factors could affect bird communities; therefore, there is a need to conduct research in developing countries.

In this review, factors that were stabilized by urbanization and that may influence the temporal dynamics of bird composition were categorized as biotic and abiotic (*Hooper et al., 2005*; *Pau et al., 2011*; *Beninde, Veith & Hochkirch, 2015*), such as food and artificial light, respectively. This classification allows us to determine the relative role of biotic or abiotic factors at the analyzed temporal scales. Therefore, the specific aims of this synthesis were to: (1) review evidence of temporal stability in biotic and abiotic factors influencing birds in urban environments across different temporal scales; and (2) assess the impact of urbanization on temporal persistence of bird species composition at different temporal scales. Finally, possible lines of research are recommended.

## SURVEY METHODOLOGY

The first step in the selection of literature for this review consisted of searching for studies on the temporal variation of bird communities and the resources they use, using keywords such as "interannual", "seasonal", "nocturnal" coupled with "urban" and "bird", and "interannual", "seasonal", "nocturnal" coupled with "fruits", "insects" "resources" and "vegetation phenology" (see Table S1). I used Google Scholar during December 2017 and
reviewed the first 300 returns ordered by relevance for each keyword, and the Scopus database in March 2018 for papers published since database inception, including those terms in the title, abstract and keywords. Moreover, I received weekly Google scholar alerts about papers published with the terms "urban" and "birds". I included only articles that compared temporal variation of bird composition or resources between different levels of urbanization (urban vs suburban, urban vs rural, etc.) or different levels or environmental conditions related to urbanization (impervious cover, light and noise levels). I did not include studies that analyzed long-term dynamics of bird communities in urban parks or suburban areas that also underwent changes in local habitat structure and landscape composition (for example, *Walcott, 1974*; *Recher & Serventy, 1991*). In the case of nocturnal activity of diurnal birds, I took into account cases of birds singing, feeding or doing both activities.

## TEMPORAL STABILIZATION OF RESOURCES AND HABITATS

### Diurnal scale

#### Abiotic factors

Undoubtedly, artificial lighting is the most notable factor altering the natural day-night cycle in urban areas, exerting negative effects on wildlife (*Navara & Nelson, 2007*). For example, Black-tailed godwits (*Limosa limosa*) avoided illuminated areas when selecting nest sites (*Longcore & Rich, 2004*). However, some bird species may take advantage by extending their feeding times (*Deviche & Davies, 2014*; see examples below). Global maps indicate that the areas most impacted by artificial lighting are the most urbanized ones, such as east North America and Europe (*Cinzano, Falchi & Elvidge, 2001*; *Longcore & Rich, 2004*). The greatest increase of light pollution from 1992 to 2012 occurred in Mediterranean-climate ecosystems and temperate ecorregions (*Bennie et al., 2015*). Within cities, artificial lighting increases with rising urbanization level (*Kyba et al., 2012*; *Hale et al., 2013*; *Katz & Levin, 2016*), being more intense in commercial (*Lim et al., 2018*; *Ma, 2018*) than in residential areas. In Berlin, most of zenith directed light comes from streets, including direct and scattered lights from lamps, automobile headlights, advertising lights located in the street area and, to some extent, light from building facades (*Kuechly et al., 2012*). The next land use with high light emissions was the commercial/industrial/service and public service area. *Zheng et al. (2018)* found that main roads, commercial and institutional areas were brightly lit in Hangzhou (China), whereas residential, industrial and agricultural areas were dark at night. In Flagstaff (USA), 33% of uplight was found to arise from sports lighting; when sport lighting was off, commercial and industrial lighting accounted for 62% (*Luginbuhl et al., 2009*). In Reykjavik (Island), almost 50% of artificial light at night came from street lights (*Hiscocks & Gudmundsson, 2010*).

The urban heat island phenomenon may impact the day-night variation of temperature. *Lazzarini et al. (2015)* found that day-night difference of land surface temperature (LST) was lower in urban than in natural areas of desert cities in North America, Africa and the Middle East.

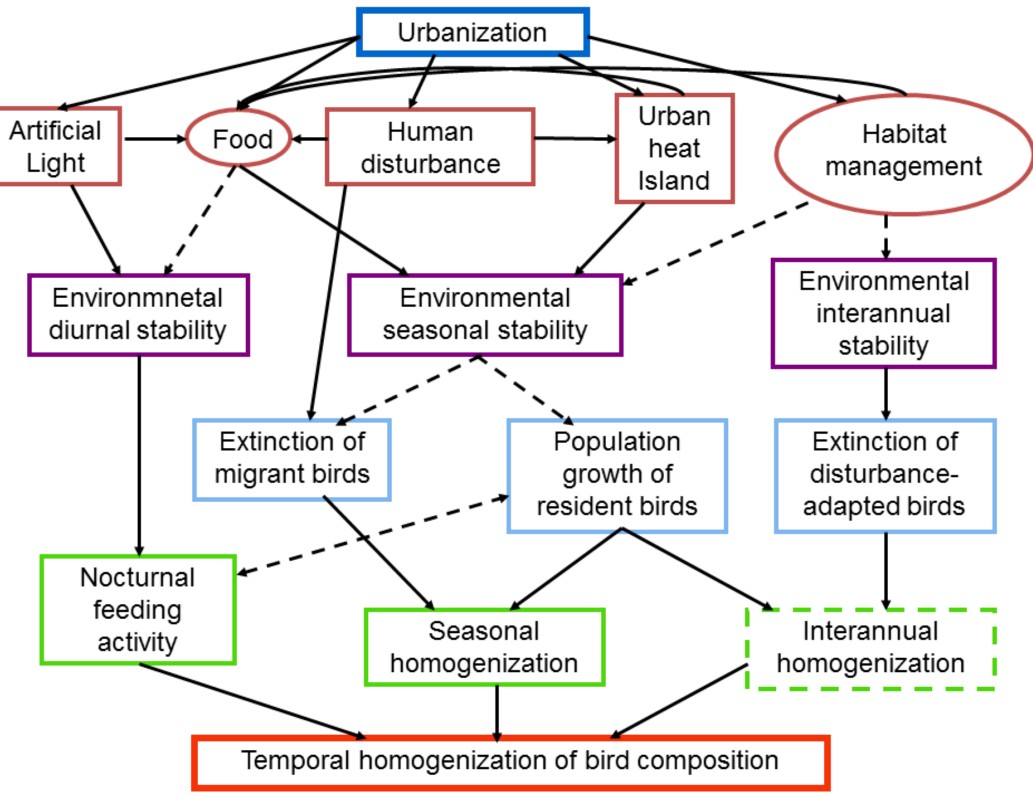

**Figure 1 Schematic diagram of intervening factors and possible mechanisms influencing the temporal dynamics of bird composition in urban areas.** Urbanization promotes the temporal stabilization of several environmental characteristics such as light, food, temperature and habitat structure (red boxes and circles). Human disturbance, such as the passing of pedestrians may provide food for birds, whereas the passing of cars may increase the urban heat island. Factors in circles may be strongly affected by the socioeconomic characteristics of citizens. The environmental stabilization is produced at different temporal scales (purple boxes), which in turn promote the temporal homogenization of bird communities. Dashed lines and boxes indicate factors and processes that require further research

### Biotic factors

Food availability at night is another factor promoting nocturnal activity of diurnal bird species (Fig. 1). Artificial lighting attracts invertebrates, which are more abundant under street lights than in patches between lights (*Scanlon & Petit, 2008*; *Davies, Bennie & Gaston, 2012*) (but see *Farnworth et al., 2018*). By attracting invertebrates, street lights provide potential food resources for insectivorous birds that extend their activity time after sunset (Fig. 1). However, the attraction of invertebrates by lamps depends on the type of light (*Eisenbeis & Hänel, 2009*; *Longcore et al., 2015*). *Eisenbeis & Hänel (2009)* found that mercury lamps attracted more insects than sodium lamps. In aquatic environments, artificial light at night has been shown to alter the activity of diurnal fish species, allowing a constant activity of the Baunco fish (*Girella laevifrons*) through the day and night (*Pulgar et al., 2018*). Moreover, diurnal raptors may exploit alternative food resources available at night at sites with artificial light, such as bats (*Mikula et al., 2016*).

## Seasonal scale
### Abiotic factors

Abiotic factors altered by urbanization were mainly temperature and wind. Temperature may influence birds directly via the urban heat island phenomenon (*Rizwan, Dennis & Chunho, 2008*), by increasing temperatures during winter and favoring bird presence (*Leston & Rodewald, 2006*). The urban heat island may have an indirect effect on seasonal stabilization of resources used by birds by favoring an extended growing season of vegetation (see below Biotic factors).

A number of studies analyzed the seasonal variation of temperature along urbanization gradients. However, most of them were concentrated in China, especially in Beijing (Table S2). The majority of studies were concentrated between 2010 and 2018 (22 of 24 studies), and the most widely used remote sensors were the Moderate Resolution Imaging Spectroradiometer (MODIS) and Landsat 5 and 7, although both have contrasting spatial resolutions (1,000 and 60–120 m, respectively). These remote sensors measured the land surface temperature (LST). In general, studies found that urban areas had a higher seasonal LST variation than rural or natural areas. Other studies that compared urban with desert or sand dune areas showed a smaller seasonal change of LST in urban areas. This heterogeneity in results suggest that seasonal fluctuations of LST associated with urbanization are context-dependent. Even within a given study area, results vary for different land uses compared to the urban area. For example, *Meng et al. (2009)* found that the seasonal change of LST was higher in the urban area than in the forest, but seasonal change of LST was bigger in the grassland than in the urban area.

Seasonal fluctuation of wind intensity may be altered by urbanization. A study in Phoenix (USA) showed that in urban areas, wind speed was significantly reduced, contributing to increased plant growth and accumulated biomass (*Bang, Sabo & Faeth, 2010*). This increase in plant biomass can have positive impacts on higher trophic levels, such as birds. Similarly, a reduced wind speed may have direct positive effects on the reproductive success of birds by reducing the probabilities that eggs and nestlings fall from the nest (L Leveau, pers. obs., 2014).

### Biotic factors

Biotic factors stabilized by urbanization at the seasonal scale included vegetation, abundance of arthropods and food supplied by humans. Abiotic factors, such as temperature and irrigation, also affected the extended growing season of vegetation.

Evidence of altered seasonal patterns of resources came mainly from data of leaf and flowering phenology and vegetation indices obtained from remote sensors; those data are mainly indicators of the net primary productivity. Most studies were conducted in the Northern Hemisphere, mainly in USA and China; and were concentrated in the 2010–2018 period (10 out of 12 studies) (Table S3). The most widely used remote sensor was MODIS, which produces the MOD13Q1 database that contains images of the Normalized Difference Vegetation Index (NDVI) and the Enhanced Vegetation Index (EVI). These images are produced every 16 days and are freely available. Several studies found an extended growing period of vegetation in urban environments, associated with the increase in temperature
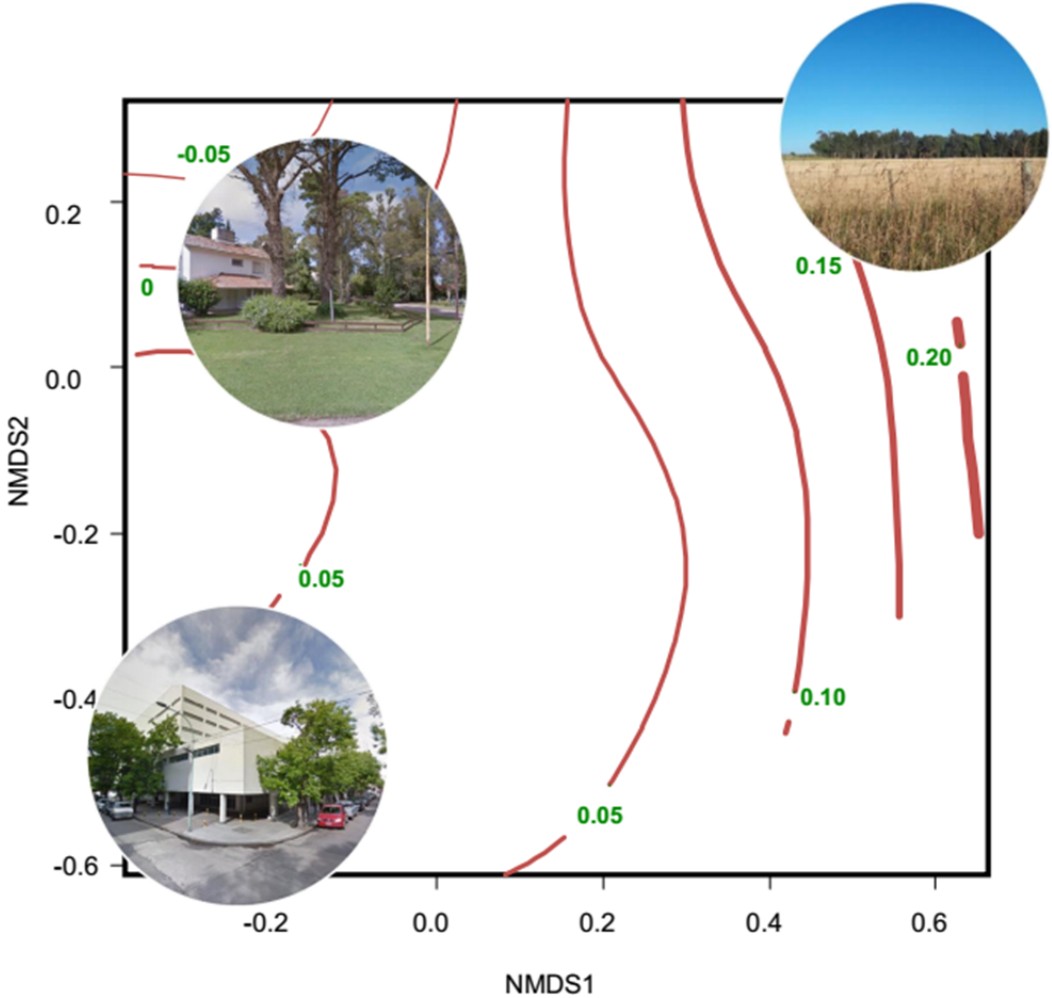

**Figure 2** **Non-metric multidimensional scaling showing the relationship between habitat types and the seasonal change of NDVI.** The ordination was constructed with a matrix of percent cover of land cover types (vegetation and impervious surfaces) and the number of high trees (<5 m, Tree_more5m) and low trees (<5 m, Tree_less5m) as columns, and each sampling unit as rows. Then, a Gower dissimilitude distance was calculated. Finally, a surface of the seasonal change of NDVI (red lines with their values) was added to the ordination. The seasonal change is the mean NDVI of spring–summer minus the mean NDVI of fall-winter. Thicker lines indicate higher values of the seasonal change of NDVI. Modified from *Leveau, Isla & Bellocq (2018)*

induced by the urban heat island (*Imhoff et al., 2000*; *Luo et al., 2007*; *Buyantuyev & Wu, 2012*; *Zhou et al., 2016*). Other studies showed a smaller seasonal change of primary productivity in urban areas than in natural and rural areas (*Coops, Wulder & Iwanicka, 2009*; *Coetzee & Chown, 2016*; *Leveau, Isla & Bellocq, 2018*). Along the urban-rural gradient of Mar del Plata (Argentina), the lowest seasonal change of primary productivity was related to high percentage cover of buildings and vegetation typical of residential houses (lawn, shrubs and trees), whereas the greatest seasonal change of primary productivity was related to non-managed herbaceous vegetation and crops (Fig. 2).

*Neil & Wu (2006)* conducted a review about the effect of urbanization on plant phenology and found that spring-blooming plants tended to bloom earlier in a variety of ecosystems in North America, Europe and China. Moreover, *ffrench Constant et al. (2016)* related budburst data from four deciduous trees to night-time lighting using satellite data as well as spring temperature. The authors found that tree budburst occurred 7.5 days earlier in brighter areas, which indicated the presence of urban areas in United Kingdom. In Florence (Italy), light exposure from luminaires extended the greening of leaves by nearly 20 days (*Massetti, 2018*). Due to a warmer microclimate, plants had less frost damage in urban areas of Phoenix (USA) than in the desert and, therefore, green leaves were more persistent in urban areas (*Bang, Faeth & Sabo, 2012*). In the same city, plants flowered for a longer period than their desert counterparts, promoting a higher abundance of pollinators in urban areas (*Neil et al., 2014*). In South Korea, four species of plants extended their flowering period in the most urbanized areas (*Jeong et al., 2011*). *Comber & Brunsdon (2015)* found that the first flowering occurred earlier in urban areas of the United Kingdom than in other land uses, but the effect of urbanization varied spatially, suggesting the need for spatially explicit analysis. *Gorton, Moeller & Tiffin (2018)* found that the plant *Ambrosia artemissifolia* flowered earlier in urban than in rural habitats of Minneapolis (USA), and attributed this asynchronous flowering to genetic differences between populations. On the other hand, *Davis, Major & Taylor (2016)* found that native trees in Sydney (Australia) flowered longer in streets than in remnant and continuous forests, providing more food resources to nectarivorous birds. Nectar and fruits may be available during seasons when they would be naturally absent or scarce due to the planting of ornamental trees (*Williams & Karl, 1996*; *Corlett, 2005*; *Leston & Rodewald, 2006*; *Williams et al., 2006*; *Montaldo, 1984*; *Leveau, 2008*; *Leveau & Leveau, 2011*; *Davis, Major & Taylor, 2015*; *Gray & Van Heezik, 2016*) (Fig. 1). On the other hand, a couple of studies found a greater seasonal change of primary productivity and a shorter growing season in an urban area than in a forest (*Chang et al., 2011*; *Mishra & Chaudhuri, 2015*; Table S3), suggesting that urbanization impacts on seasonal dynamics of primary productivity are biome-dependent.

Irrigation, fertilizer application and pruning, and the planting of perennial tree species such as *Pinus* sp. and *Eucalyptus* sp. may stabilize the seasonal dynamics of primary productivity in urban areas compared to agricultural and natural areas (*González-García & Sal, 2008*; *Loram et al., 2011*; *Buyantuyev & Wu, 2012*; *Leong & Roderick, 2015*) (Fig. 1). The longer period of growing and stability of vegetation in cities may impact other organisms, such as invertebrates, especially if vegetation exerts bottom-up control on them (*Leong & Roderick, 2015*). Therefore, there would be more food resources available to birds throughout the year.

Studies that analyzed the seasonal variation of arthropod abundance and frequency along urbanization gradients were mainly concentrated in the 2010–2018 period (80%, $n = 20$) and were equally distributed in North America, Europe, Asia, and South America (Table S4). In general, there was a lower seasonal change in abundance of arthropods in urban than in rural or natural areas. Some studies showed a lower variation in intermediate levels of urbanization and industrial areas (*McIntyre et al., 2001*; *Mulieri et al., 2011*). Other

studies found a similar variation between urbanization levels (*Carbajo et al., 2004*) or a higher variation in urban areas (*Baruah & Dutta, 2013*).

The provision of bird feeders with grains or nectar is an important stabilizing factor for birds in urban areas, but they are mainly distributed in cities of developed countries (*Jones & James Reynolds, 2008*) (Fig. 1). In Europe, bird feeding was more frequent in Scandinavian countries and Ireland (*Thompson, Greenwood & Greenaway, 1993*). Compared with rural areas, the number and type of food provided by humans is higher in urban areas (*Tryjanowski et al., 2015*). While urban areas are characterized by large numbers of bird tables and human food provided on the ground, rural areas showed a more frequent use of pig fat and/or skin sometimes mixed with some seeds and prepared as a block or ball (*Tryjanowski et al., 2015*). Within each city, bird feeding was more frequent in small parks (*Thompson, Greenwood & Greenaway, 1993*; *Gaston et al., 2007*) and in feeders located at high altitude (*Thompson, Greenwood & Greenaway, 1993*). Moreover, several authors showed that bird feeding was related to socioeconomic and demographic factors (*Lepczyk, Mertig & Liu, 2004*; *Arizmendi et al., 2008*; *Fuller et al., 2008*; *Davies et al., 2012*; *Sonne et al., 2016*; *Ramírez-Segura, 2016*). For example, the provision of seeds and artificial nectar is related to high socioeconomic levels (*Davies et al., 2012*; *Ramírez-Segura, 2016*). Over the year, food provided by humans was most frequent during winter, although food composition did not vary (*Cowie & Hinsley, 1988*). Finally, the presence of landfills may be a constant source of food for gulls, ibis and raptors throughout the year (*Yorio & Giaccardi, 2002*; *Martin, French & Major, 2010*; *Marateo et al., 2013*; *Oro et al., 2013*; *Plaza & Lambertucci, 2017*).

## Interannual scale
### Abiotic factors
Natural disturbances that strongly impact vegetation structure and bird communities, such as hurricanes, tornados or fire (*Liu et al., 1997*; *Waide, 1991*; *Yaukey, 2012*), may be reduced or suppressed in urban areas. *Kingfield & De Beurs (2017)* found that tornado impact on NDVI was lowest in the most densely urbanized areas in USA. Fire activity was minimal at the edge of urban areas in the Sierras chicas (Argentina) and North Carolina (USA) (*Argañaraz et al., 2015*; *Pickens et al., 2017*). However, *Branoff (2018)* found no impacts of urbanization on mangroves vegetation resistance to hurricanes. On the other hand, floods may be increased by urbanization (*Hollis, 1975*; *Konrad, 2003*; *Jacobson, 2011*).

Case studies that analyzed the interannual variation of temperature in urban areas are scarce. A study from China showed that interannual variation in LST was higher in urban areas than in forests (*Zhao et al., 2009*).

### Biotic factors
A few studies from North America provided remote sensing data and suggested that urban areas have a lower interannual variability of primary productivity than natural areas (*Shochat et al., 2004*; *Leong & Roderick, 2015*). Therefore, habitat structure and food resources for birds may be temporally more stable in urban than in rural areas. In addition, *Bang, Faeth & Sabo (2012)* found that the impact of interannual variability of precipitation had greater effects on plant growth and arthropod abundance in desert than in urban areas.

Activities such as maintaining vegetation or using fertilizers and pesticides may stabilize the interannual habitat structure and primary productivity of residential areas (*Lepczyk, Mertig & Liu, 2004*) (Fig. 1). Furthermore, using and maintaining nest boxes in backyards may stabilize the presence of hole-nesting bird species (*Davies et al., 2009*; *Duckworth, 2014*).

However, several authors argue that urban areas may be unstable in the long term. Physical, social and vegetation changes associated with advances in urban technology, urban decay, urban redevelopment and socioeconomic fluctuations may influence biotic communities (*Shaw, Chamberlain & Evans, 2008*; *Luck, Smallbone & O'Brien, 2009*; *Hulme-Beaman et al., 2016*). On the other hand, the use of bird feeders by households may change considerably in the long term, affecting the occurrence of bird species in backyards (*Chamberlain et al., 2005*). Finally, the colonization of cities by predators, such as birds of prey or crows (*Rutz, 2008*; *Tomiałojć, 2011*), may promote directional changes in bird composition by causing population declines in their prey species.

## TEMPORAL STABILIZATION OF BIRD COMPOSITION

### Diurnal scale

Although there are many records of nocturnal activity by diurnal birds in urban areas (*Sick & Teixeira, 1981*; *Negro et al., 2000*; *Rejt, 2004*; *DeCandido & Allen, 2006*; *Mikula, Hromada & Tryjanowski, 2013*), only a few studies related bird activity to environmental variables or urban attributes (Table 1). All the studies were conducted in the Northern Hemisphere, mainly in Europe, and nine out of 11 studies were performed in the 2010–2018 period. Six studies were focused on singing behavior, and the most extensively studied species was the Blackbird (*Turdus merula*). In general, most studies found more bird activity at night in areas with more artificial light. Other factors such as day length and cloud cover may influence bird activity at night (*Russ, Rüger & Klenke, 2015*). On the other hand, a couple of studies showed that anthropogenic noise can be the cause of earlier dawn song in birds (*Fuller, Warren & Gaston, 2007*; *Arroyo-Solís et al., 2013*).

The stabilization of bird composition between day and night can also be promoted by the extinction of nocturnal species. For example, *Weaving et al. (2011)* found that two of the three studied nocturnal species showed negative responses to urbanization in Melbourne, Australia. However, *Bosakowski & Smith (1997)* found negative effects of urbanization only on the Barred Owl (*Strix varia*), whereas the Great Horned Owl (*Bubo virginianus*) and the Eastern Screech Owl (*Otus asio*) did not exhibit responses to urbanization.

### Seasonal scale

Studies analyzing seasonal variation of bird composition along urbanization gradients are scarce and were conducted in the Southern and Northern Hemisphere, and generally performed during the 2010–2018 period. All studies showed negative effects of urbanization on seasonality of bird composition (Table 2). Most of the studies compared seasonal dynamics between habitat types, except for two studies that related seasonal changes of community composition to impervious surface cover and other vegetation characteristics (*Leveau, Isla & Bellocq, 2015*; *Leveau & Leveau, 2016*). Many examples showed loss of

**Table 1** Summary of studies assessing the nocturnal activity of birds, such as feeding (A), singing (B) or both (C) in relation to different artificial light intensities.

| Species | Main results | Location | Source |
|---|---|---|---|
| A- Feeding | | | |
| *Turdus merula* | Birds captured in the city started their activity earlier than rural birds | Munich, Germany | *Dominoni et al. (2013b)* |
| *Mimus polyglottos* | Birds fed nestlings after sunset in areas with more artificial light | Gainesville, USA | *Stracey, Wynn & Robinson (2014)* |
| *Turdus merula* | Birds foraged after sunset in areas with more artificial light | Leipzig, Germany | *Russ, Rüger & Klenke (2015)* |
| *Prunella modularis, Erithacus rubecula, Turdus merula, Cyanistes caeruleus, Parus major, Pica pica, Passer domesticus, Fringilla coelebs, Carduelis chloris* | Birds delayed the onset of foraging in gardens surrounded by more urbanization | UK | *Ockendon et al. (2009)* |
| *Streptopelia decaocto* | Birds started earlier the onset of foraging in gardens surrounded by more urbanization | UK | *Ockendon et al. (2009)* |
| *Turdus merula, Erithacus rubecula, Cyanistes caeruleus, Prunella modularis, Parus major, Fringilla coelebs, Periparus ater, Passer domesticus, Caduelis carduelis, Columba palumbus* | Birds delayed the onset of foraging in gardens with more artificial light | UK | *Clewley et al. (2016)* |
| B- Singing | | | |
| *Columba palumbus* | Calling activity was not affected by the distance to artificial light source | Greifswald, Germany | *Böhm et al. (2016)* |
| *Turdus migratorius* | Birds sing at night in areas with large amount of artificial light | Eastern USA | *Miller (2006)* |
| *Turdus merula, Erithacus rubecula, Parus major, Cyanistes caeruleus.* | Birds started singing earlier close to artificial lights | Viena, Austria | *Kempenaers et al. (2010)* |
| *Fringilla coelebs* | Birds did not start singing earlier close to artificial lights | Viena, Austria | *Kempenaers et al. (2010)* |
| *Turdus merula, Fringilla coelebs, Parus major, Cyanistes caeruleus, Erithacus rubecula* | Birds started singing earlier close to artificial lights | Oulu, Finland; Starnberg, Germany; Granada, Spain | *Da Silva & Kempenaers (2017)* |
| C - Feeding and singing | | | |
| *Pyrocephalus rubinus* | Birds sang and foraged at night in areas with more artificial light | Morelia, Mexico | *MacGregor-Fors et al. (2011)* |
| *Erithacus rubecula* | Birds sang and foraged at night close to artificial light | Bergen, Norway | *Byrkjedal, Lislevand & Vogler (2012)* |

migratory behaviour in bird species colonizing urban environments: (a) the Merlin (*Falco columbarius*, *Warkentin, James & Oliphant, 1990*); (b) the European Robin (*Erithacus rubecula*, *Adriaensen & Dhondt, 1990*); (c) the Dark-eyed Junco (*Junco hyemalis*, *Yeh, 2004*); (d) the House Sparrow (*Passer domesticus*, *Anderson, 2006*); and (e) the European blackbird (*Turdus merula*, *Partecke & Gwinner, 2007*; *Møller et al., 2014*). However, supplementary food during winter has been associated with northward winter migration of the Eurasian Blackcap (*Sylvia atricapilla*) in Europe (*Plummer et al., 2015*).

**Table 2** Summary of studies that compared the seasonal change of bird composition along urbanization gradients.

| Location | Main results | Periods compared | Source |
| --- | --- | --- | --- |
| Montpellier, France | The lowest seasonal turnover was shown in the most urbanized areas, but residential areas had lowest | Winter-Spring | *Caula, Marty & Martin (2008)* |
| North America | The lowest seasonal change was shown in urban habitats; there was a greater effect in the west than in east NA | Monthly | *La Sorte, Tingley & Hurlbert (2014)* |
| Mar del Plata, Argentina | The lowest seasonal change was shown in areas with more impervious cover | Breeding-Nonbreeding | *Leveau, Isla & Bellocq (2015)* |
| Mar del Plata, Argentina | The lowest seasonal change was shown in urban parks of the urban center | Breeding-Nonbreeding | *Leveau & Leveau (2016)* |

**Table 3** Summary of studies that compared the interannual change of bird composition along urbanization gradients.

| Location | Main result | Extent | Length | Source |
| --- | --- | --- | --- | --- |
| Brisbane, Australia | Suburban habitats had a more dynamic composition than bushland habitats | Suburban-Natural | 15 years | *Catterall et al. (2010)* |
| Kraków, Poland | There was a positive effect of artificial light and urban infrastructure on winter community stabilization | Urban-Natural | 2 years | *Ciach & Fröhlich (2017)* |
| South-east North America | No effect of urbanization on species turnover | Urban–Rural | 25 year | *Barrett, Romagosa & Williams (2008)* |
| Finland | There was a more stable composition in the more populated urban areas | Urban areas of different sizes | 8 years | *Suhonen et al. (2009)* |
| Rovaniemi, Finland | There was a more stable composition near to the urban centre | Suburban–Exurban | 5 years | *Jokimäki & Kaisanlahti-Jokimäki (2012a)* |
| Mar del Plata, Argentina | There was a more stable composition in the urban centre, more stable during the breeding season | Urban-Periurban | 3 years | *Leveau & Leveau (2012)* |
| Mar del Plata, Argentina | There was a more stable composition in the more urbanized sites | Urban–Rural | 3 years | *Leveau, Isla & Bellocq (2015)* |
| Phoenix, USA | There was a more stable composition in the more urbanized riparian sites | Riparian sites along an urbanization gradient | 12 years | *Banville et al. (2017)* |

## Interannual scale

Most of the eight studies analyzed were conducted in the Northern Hemisphere, and during the 2010–2018 period (six out of eight studies). Five studies were long term (>5 years). In general, studies showed that urbanization stabilized the interannual variation of community composition (Table 3). However, two studies that spanned the longest term found null effects of urbanization or opposite trends than expected (*Barrett, Romagosa & Williams, 2008*; *Catterall et al., 2010*). In particular, *Catterall et al. (2010)* found population declines of urban-associated species, such as the House Sparrow or European Starling (*Sturnus vulgaris*) in suburban habitats compared with bushland. Most of studies related community composition stability to urban attributes, such as impervious cover or human density.

## POSSIBLE MECHANISMS

The role of stabilization of biotic and abiotic factors seems to vary at the different temporal scales. At the diurnal scale, stabilization of abiotic conditions such as light seems to be the most relevant, inducing nocturnal activity of birds. On the other hand, artificial light can increase food availability for nocturnal birds, especially insectivores. However, in the case of species that feed on human food waste or earthworms, such as *Columba livia* or *Turdus* sp., food resource does not seem to be a limiting factor at night. Moreover, without artificial light, it is difficult that diurnal species could find food. At the seasonal and interannual scales, the stabilization of biotic factors seems to be the dominant determinant of bird composition stabilization. However, this stabilization of biotic factors is usually mediated by the effect of abiotic factors, such as water or the urban heat island phenomenon during winter.

### Diurnal homogenization of bird composition

Studies suggest that nocturnal activity is mainly regulated by artificial light, which alters melatonin secretion (*Dominoni et al., 2013a*). Melatonin is a hormone related to the biological rhythm in animals (*Jones et al., 2015*). Daylength may be a significant factor, as birds presented the greatest night activity on shortest days (*Russ, Rüger & Klenke, 2015*; *Dominoni & Partecke, 2015*). Meteorological factors such as temperature and cloud cover may influence the nocturnal activity of birds (*Russ, Rüger & Klenke, 2015*; *Dominoni et al., 2014*). Bird density was positively related to the degree of nocturnal activity, suggesting a role of intraspecific competition (*Russ, Rüger & Klenke, 2015*; *Dominoni et al., 2013b*). Alternatively, a greater bird density may promote a greater probability of appearance of nocturnal feeding (Fig. 1). Finally, the importance of food availability at night was suggested, but it was very little explored (*Dominoni et al., 2014*). However, other authors postulated that birds may have lower energetic demands during winter and may delay the start of foraging activity as a response to the urban heat island phenomenon (*Ockendon et al., 2009*; *Clewley et al., 2016*).

### Seasonal homogenization of bird composition

The lower seasonal variation of bird communities in urban than in non-urban areas may be related to the extinction of migratory species and to the lower seasonal variation of resident species (*Pennington, Hansel & Blair, 2008*; *Jokimäki & Kaisanlahti-Jokimäki, 2012b*). Migratory species arrive at a site to exploit surplus resources not used by year-round residents (*MacArthur, 1959*; *Hurlbert & Haskell, 2003*). If urban areas were characterized by a stabilization of resources, the surplus provided to migratory species would be diminished (Fig. 1). Nevertheless, urban areas may be characterized by a low amount of food resources to migrants, especially those that feed on insects (*Faeth et al., 2005*; *Teglhøj, 2017*). Moreover, human disturbance such as pedestrian and car traffic, and nest parasitism may be especially negative to migratory birds (*Burger & Gochfeld, 1991*; *Zhou & Chu, 2012*; *Rodewald & Shustack, 2008*) (Fig. 1). In seasonally temperate areas, constant food availability to omnivorous and insectivorous species throughout the year may stabilize their temporal variation relative to populations in rural or natural

areas. For instance, the African woolly-necked stork (*Ciconia microscelis*) were provided supplementary food by humans year-round in KwaZulu-Natal (South Africa), and humans were motivated by pleasure (*Thabethe & Downs, 2018*). Silvereyes (*Zosterops lateralis*) in Dunedin (New Zealand) complemented their seasonal foraging on insects by using exotic and native trees (*Waite et al., 2013*); the exotic English oak (*Quercus robur*) had the highest abundance of arthropods. The advance of reproductive phenology in the Abert 's Towhee (*Melozone aberti*) in Phoenix was related to predictability and limited change of food resources in urban areas (*Davies et al., 2016*).

The increased stable presence of resident species may negatively affect migratory species by interspecific competition for food and nesting places (Fig. 1). For example, House sparrows may use aerial hawking of insects, depleting resources for migratory species that exploit the same resources. House Sparrows and European Starlings nest in holes of buildings and trees, the same nest substrates as those used by several migratory swallow and swift species in the Neotropic, Neartic and Paleartic regions (*Palomino & Carrascal, 2006*; *Leveau, Isla & Bellocq, 2015*; *Chantler & Boesman, 2018*). Finally, supplementary feeding during winter may generate a surplus of resources, favouring the arrival of winter migrants (*Plummer et al., 2015*), and therefore promoting an opposite process to seasonal homogenization of bird communities.

On the other hand, quantity and quality of food, artificial light and the urban heat island phenomenon may affect the physiology of bird individuals, altering their phenology by lengthening the breeding season (*Deviche & Davies, 2014*). A continental study across Europe showed that several species had a longer singing period (as a proxy of the breeding season) in urban than in rural areas (*Møller et al., 2015*). Moreover, this effect of urbanization was higher in species that colonized cities long before other species and in the biggest cities, due to a target effect and a lower risk of extinction (mechanism of island biogeography *Møller et al., 2015*). It is noteworthy that this longer breeding season in urban areas was also observed in migratory species, despite the negative effects of urbanization on this group of species (see references above).

## Interannual homogenization of bird composition

Interannual composition stability in highly urbanized areas is probably driven by the high densities that some species reach there, favoured by several factors such as a constant food supply, habitat stability and a favourable microclimate during winter (Fig. 1). A high dominance of a few species, typically the House Sparrow and the Rock Dove, may diminish their extinction probabilities at local scales, at least in the short and mid-term (between 2 and 10 years). However, at longer temporal scales (>10 years), biotic instabilities may occur due to strong competitive interactions between species (*DeAngelis & Waterhouse, 1987*). *Shochat et al. (2010)* showed that dominant bird species were highly efficient foragers that leave scarce resources to subordinate species, probably leading to their exclusion in the long term. On the other hand, bird community fluctuations are probably more governed by environmental stochasticities in rural or natural than in urban areas, due to natural disturbances or climatic fluctuations, which affect the persistence of rare species, leading to increased temporal variation of bird composition (*Collins, 2000*; *Sasaki & Lauenroth, 2011*).

## EFFECTS ON THE TROPHIC DYNAMICS OF PREDATORS

The dampening in biotic resources and bird species composition in urban areas may impact the trophic dynamics of vertebrate predators, such as cats or raptors. For example, free-ranging cats had less marked seasonal variations of diet composition in urban than in rural areas (*Krauze-Gryz, Zmihorski & Gryz, 2017*). Urban Black vultures (*Coragyps atratus*) presented a uniform use of carrion throughout the year, whereas rural vultures showed a fluctuation in diet composition from arthropods to carrion (*Ballejo & De Santis, 2013*). Finally, the seasonal variation of diet composition in the Tawny owl (*Strix aluco*) was lower in the city of Warsaw than in a nearby forest, mainly due to predation on the resident House sparrow (*Passer domesticus*) (*Goszczyński et al., 1993*).

## THE NEED TO DIFFERENTIATE SPATIAL AND TEMPORAL HOMOGENIZATION OF AVIFAUNAS

In my opinion, spatial and temporal homogenization acts at different spatial scales and has different underlying processes. On the one hand, spatial homogenization is the increase of taxonomic similarity between two or more biotas over a specified time interval (*Olden & Rooney, 2006*). The main process is the extinction of native species and the colonization of widespread species. On the other hand, temporal homogenization is the increase of taxonomic similarity in a given biota over time. This phenomenon occurs at several temporal scales, as mentioned above. Underlying processes are the extinction of migratory or disturbance-dependent species and the colonization of new temporal niches (e.g., the night; *Hut et al., 2012*).

Are both processes simultaneous? The spatial homogenization of bird composition promoted by urbanization seems to be scale-dependent. While at a global scale or when comparing different biomes, there is a higher taxonomic similarity in the most urbanized areas than in less urbanized and non-urban areas (*Clergeau, Jokimäki & Savard, 2001*; *Leveau et al., 2017*), at the regional scale avifaunas of highly urbanized areas seem to be as heterogeneous as suburban avifaunas (*Leveau, Jokimäki & Kaisanlahti-Jokimäki, 2017*). On the other hand, temporal homogenization of avifaunas can be considered at different spatial scales, from the local to the global scale. For example, seasonal stabilization of bird composition was detected at local scales in Argentina (*Leveau, Isla & Bellocq, 2015*; *Leveau & Leveau, 2016*) and among different biomes in North America (*La Sorte, Tingley & Hurlbert, 2014*). The nocturnal activity of Rock Doves in urban centres is a global phenomenon (*Luniak, 2004*; L Leveau, 2016, unpublished data).

## CONCLUSIONS AND FUTURE DIRECTIONS

Published evidence suggests that urban areas dampen the natural cycles at several temporal scales. The stabilization of biotic and abiotic factors, such as light, temperature, food and habitat structure, are desynchronized from natural diurnal, seasonal and interannual cycles. These changes induced by urbanization are expected to influence the temporal dynamics of bird composition. In fact, the reviewed literature showed that bird composition was

temporally most stable in the most urbanized areas, leading to a temporal homogenization of bird communities (Fig. 1). As a result, urbanization promotes a decline of temporal beta diversity. However, the studies analyzed only covered taxonomic diversity, whereas other facets of biodiversity, such as functional or phylogenetic diversities, remain unexplored.

A possible consequence of resource and habitat stabilization is population growth and range expansion of bird species adapted to urban conditions. For instance, a recent study showed that daily nest survival rates of Blackbirds increased with artificial light at night (*Russ, Lučeničová & Klenke, 2017*). On the other hand, clear examples of species with range expansions associated with urban conditions are hummingbirds; for example, the Anna's Hummingbird (*Calypte anna*), the Allen's Hummingbird (*Selasphorus sasin sedentarius*) and the White-throated Hummingbird (*Leucochloris albicollis*) have expanded their range sizes and are highly associated with residential areas, which provide supplementary food resources such as nectar, year-round flowering plants and nesting sites (*Clark & Russell, 2012*; *Clark, 2017*; *Greig, Wood & Bonter, 2017*; *Weller, Kirwan & Boesman, 2017*). Psitacidae species, such as the Rose-ringed parakeet (*Psittacula krameri*) and the Monk Parakeet (*Myiopsitta monachus*) in Europe and the Rainbow Lorikeet (*Trichoglossus haematodus*) in Australia, expanded their distributions associated with food and nesting resources provided in urban areas (*Shukuroglou & Mccarthy, 2006*; *Strubbe & Matthysen, 2009*; *Clergeau & Vergnes, 2011*). Despite these findings, there are many research gaps that need to be filled.

Studies concerning the nocturnal activity of diurnal species were focused on the analysis of intervening factors and experimentation, but research was exhaustive only on the European Blackbird. It is noteworthy that other globally distributed species with nocturnal activity such as the Rock Dove have still not been studied (*Luniak, 2004*). This is an interesting model species to explore environmental conditions affecting nocturnal activity.

Although there are no studies that compare bird composition similarity between day and night, in those studies reporting nocturnal activity of some diurnal species, I assumed that those species also had diurnal activity that was not mentioned. In fact, several studies showed this pattern (*Byrkjedal, Lislevand & Vogler, 2012*; *MacGregor-Fors et al., 2011*; *Russ, Rüger & Klenke, 2015*). Therefore, in those urbanized areas, species similarity between day and night is higher than in areas where there is no nocturnal activity of typical diurnal birds. However, there is a need to formally compare the similarity in composition between day and night along urbanization gradients.

Comparisons of seasonal and interannual compositional stability between urbanization levels have been conducted for different countries (Table 1); however, studies that relate compositional stability to temporal environmental variation are scarce. A recent study showed that seasonal change of primary productivity was directly related to seasonal change of bird composition (*Leveau, Isla & Bellocq, 2018*). In addition, experimental studies controlling or altering the temporal availability of resources and their effect on bird composition stability are needed. A relevant example is a work of *Galbraith et al. (2015)*, in which supplementary food was experimentally added in residential areas and removed after 18 months, promoting a higher temporal variation of bird composition than in control sites where food availability was not altered.

Given that wealthier householders may provide more resources for habitat management and bird feeding, leading to a higher biodiversity (the so called luxury effect, *Leong, Dunn & Trautwein, 2018*), it is expected that this type of intervention may promote an increase in temporal stabilization in bird composition. These socioeconomic contrasts may act at different spatial scales, from the local scale, comparing different residential areas within a city, to the global scale, comparing cities of developed and developing countries, or arid versus tropical biomes (*Leong, Dunn & Trautwein, 2018*).

The stabilizing role of urbanization in environmental conditions may be more or less evident, depending on the geographical location of cities. For example, desert biomes are highly variable according to interannual precipitations (*Fang et al., 2001*) and, therefore, irrigation in residential areas may exert higher effects on the stability of bird composition than irrigation in urban areas of forested biomes, which may have lower interannual fluctuations in precipitation.

As a result of higher environmental and population variability with time and habitat succession, bird communities are expected to show more variability as the time of observations of studies increases (*Bengtsson, Baillie & Lawton, 1997*). In part, our review resulted in patterns of interannual variability opposed to those expected for those studies with more than 10 years of observation period (Table 1). Therefore, studies conducted at longer temporal scales are needed (see also *Fidino & Magle, 2017*). On the other hand, although bird composition may change greatly between years, it is necessary to explore whether bird communities of urban areas undergo directional changes through time (*Collins, 2000*; *Collins, Micheli & Hartt, 2000*).

An overlooked aspect of urban environments is the possible creation of rhythms in bird communities as a response to massive movement of people between weekdays and weekends. Car traffic may promote weekly cycles of pollutants, wind speed and noise, which may cause changes in bird activity (*Shutters & Balling, 2006*; *Leveau, 2008*; *Halfwerk et al., 2011*). Along a road near Madrid, *Bautista et al. (2004)* found changes in raptor composition between weekdays and weekends. On the other hand, *Fernández-Juricic et al. (2003)* found that the presence of pedestrians in urban parks of Madrid affected the abundance of House Sparrows. *Lafferty (2001)* found a marginal difference ($P < 0.10$) of bird abundance between weekdays and weekends in sandy beaches of California and a significant interaction between weekends and seasons, suggesting that the effect of human density may depend on the period of the year and bird phenology. Therefore, we may expect significant changes in bird composition between weekdays and weekends in urban green spaces, because a massive number of people usually visit them on weekends.

The dampening of temporal variation in bird composition may be another aspect of the so-called extinction of experience (*Pyle, 1978*; *Miller, 2005*), i.e., humans are disconnected from natural cycles. It has been reported that most people liked the change of seasons (*Jauhiainen & Mönkkönen, 2005*; *Soga et al., 2016*), and *Palang et al. (2005)* recommended to pay more attention to seasonality in landscape planning. Moreover, although little explored, the seasonal variation of vegetation and animal activity may have an aesthetic value and support ecosystem services (*Dronova, 2017*). For instance, *Graves, Pearson & Turner (2018)* found that the type of cultural ecosystem services, such as birdwatching,

varied spatio-temporally along an urbanization gradient, because resident and migrant species had different responses to land covers. From a point of view of urban design, it is a paradox that urban habitats that we ignore, such as vacant lots, are more related to the particular natural processes of the region in which the city is located than those places designed in a formal way with particular proportions of lawn, shrubs and trees (*Hough, 1994*; *Kwok, 2018*). Therefore, a crucial step to restoring nature in cities is to pay attention to the natural cycles that are characteristic of the surroundings of cities.

Recently, a collaborative project that encompassed six metropolitan areas of varied climatic regions of the USA aimed to elucidate the ecological homogenization of urban areas compared to natural areas, showing convergences of soil moisture, amount of organic matter and microclimate among residential areas (*Groffman et al., 2014*; *Hall et al., 2016*). In my opinion, a step forward would be to analyze how these ecosystem properties are stabilized over time, and how this affects the temporal persistence of bird species.

## ACKNOWLEDGEMENTS

I thank the Editor, Piotr Tryjanowski, Adrian Davis, Jukka Jokimäki, Jorgelina Brasca, Paloma Garcia Orza and two anonymous reviewers for valuable comments that improved the manuscript.

### Funding

This work was partially supported by the Agencia Nacional de Promoción Científica y Tecnológica (PICT 2017-1652). There was no additional external funding received for this study. The funders had no role in study design, data collection and analysis, decision to publish, or preparation of the manuscript.

### Grant Disclosures

The following grant information was disclosed by the author:
Agencia Nacional de Promoción Científica y Tecnológica: PICT 2017-1652.

### Competing Interests

The author declares there are no competing interests.

### Author Contributions

- Lucas Matías Leveau conceived and designed the experiments, performed the experiments, analyzed the data, contributed reagents/materials/analysis tools, prepared figures and/or tables, authored or reviewed drafts of the paper, approved the final draft.

### Data Availability

This study did not generate raw data; this is a literature review.

## Supplemental Information

Supplemental information for this article can be found online at http://dx.doi.org/10.7717/peerj.6056#supplemental-information.

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
