# Peer review of "Urbanization, environmental stabilization and temporal persistence of bird species: a view from Latin America"

_PeerJ, doi:10.7717/peerj.6056_

## Round 0.1 · original submission · Major Revisions

Speaking honestly, I agree with many of the arguments made by referee number 3 and the fact that the manuscript needs a lot of work, but I personally believe that is possible to improve. Obviously, I will ask the same referee once again for the comments, and I also will try to find one more additional person with "fresh" vision of the MS.

Please respond to all the comments from all 3 reviewers.

·

Basic reporting

A few grammatical/English mistakes throughout, which need to be edited and corrected- for example L50, there are grammar mistakes. Otherwise yes, professional.

Background is fine. There is more literature that could be used, however this would moreso be for discussing and elaborating upon points. For example- you only gave a single sentence to nest boxes in urban areas. There is more literature that could be added here and discussed in much more detail- it depends on the depth you want to go. I think you've covered the basics of the temporal resource dynamics, but there are a few more papers dealing with avian homogenisation and urban adaptation/exploitation (recent ones on parrots in Australia, plus a few from South Africa (e.g. Hornbills) and South America) that could be worth adding into the discussion, if you think they add value.

Professional structure-yes

Self-contained- yes

Experimental design

Original Primary Research within Aims and Scope- yes

Research Question well defined, meaningful and fills gaps- Yes. It addresses some very interesting, relevant and applied aspects of urban ecology and diversity

Rigorous Investigation and Details Methods:
- One concern here is that you only used Google Scholar. Can you be sure that Google Scholar picked up all the relevant research that a search engine such as Web of Science would have picked up? Google Scholar is fine, however used in isolation, I'm just worried some key literature may have been missed?

-Additionally, I think that there also needs to be more detail in your methods. You said keywords used 'such as'. I think you should list all the keywords used to search. What criteria did you use for inclusion/exclusion?- only peer reviewed? English language? Published time period? And how many did you start with, exclude at each step and then end up with?

Validity of the findings

A good presentation of the current available research on temporal stability of resources and avian homogenisation, with well considered conclusions.

Additional comments

Some specific comments

L50- fix up grammar (and little things like this throughout)
LL64-65. Can you provide an example?
LL70-73: I don't quite understand this part about the from a human perspective?...
L91: can you give some examples (positive and negative)?
L112-113: is the association positive or negative?
L156-157: expand a little more- not only an important topic, but needs at least another sentence or two to explain the significance a little further
L224-226: Make it clear where you are talking about- really important as different for different countries etc
L228-295: There is also some parallel literature on nectarivorous lorikeets from Australia

Reviewer 2 ·

Basic reporting

The task of the review work is to show the weaknesses and strengths of scientific cognition and to draw conclusions important for future research directions. The latter depend on a thorough analysis of problems, depending on precisely formulated questions.

In recent years there have been several works or books summarizing some aspects of bird ecology in urban areas (Marzluff 2001, Chace and Walsh 2006, Chamberlain 2009, Lepczyk et al. 2012), however, because new data is constantly being added, a look updating the knowledge of certain areas of issues and deriving new research questions is valuable. This is the intention of the author, who attempts to summarize the knowledge on the diversity of the urban bird communities structure in terms of the stability of urban environment resources in three time aspects - short-term changes (night compared to the day), seasonal and long-term (rows 68-70). Generally, the idea of the work is interesting and worth continuing, because the author intends to review the state of knowledge regarding the avifauna of cities of different continents, geographical regions, areas located at different latitudes, which could lead to valuable comparisons and generalizations.

In my opinion, the weakness of the research assumptions (rows 68-70 and 74-77) is the small precision of the questions (study aims). The intention contained in the title and being the formulated aims of the work is not exactly as precise as it is for me. There is no description what aspects of stability of the bird communities the author analyzes, or only species composition ?, or also the population size? or diversity? – after all, in relation to the questions asked, the content of the result chapters summarizing the data contained in the various works should be constructed. Chapter 4 (row 165) suggests that only the species composition, but in terms of diurnal variation (chapter 4.1), the author presents and analyzes data on various aspects of the birds activity (rows 167-173), not the diversity of the species composition of communities. It is difficult to explicitly agree with the validity of the concept of the author analyzing the changes of avifauna compositions (or homogeneity) between the night and day. Similarly, in the aspect of seasonal changes (chapter 4.2), it undertakes an analysis of comparing the species composition of urban and suburban areas (rows 186-187, Table 1B) – there is a lack of a clear division into various communities – breeding avifauna, post-breeding avifauna, and finally, in certain latitudes, avifauna of wintering period. Various ecological factors may affect the quantitative and qualitative structure of birds communities. In the aspect of indicating greater settling of species and the emergence of partial migration an unnecessary reference to Fig. 3 (rows 184-185), which presents several photographs of selected species. These species could only be mentioned in this paragraph. This figure does not bring any relevant information to work.

Ecological factors the author divided in few types: the role of lighting, food, temperature and urban space management (Figure 1). Presentation of the state of knowledge about these constituents of the urban environment is quite chaotic (chapters 3.1-3.3). Most researchers in the division of environmental factors pay attention to the coverage of vegetation, microclimate, the presence of additional food sources, changed predation pressure, environmental quality (pollution, also lighting as pollutions type). Personally, in such a division context, data should be analyzed, but even with such a division used by the author (fig. 1), the information in the chapters 3.1-3.3 is missing, which we know about the differentiation of these factors, and what the author describes as "resource stabilization". There is a lack of a similar method of analysis that the author adopted when trying to describe the diversity of species composition, presenting the data in Table 1.
Figure 1 showing the scheme of impact or linking various ecological factors and variability of the composition of birds communities is interesting, but should result from the analysis.

Experimental design

Another aspect is the completeness of the analyzed data – I am not sure if the method used (rows 78-86) ensured the proper selection of papers for analysis. It should be only a starting point, because it depends on the introduction of digital work and defining key words. After selecting the papers, the source publications could be reached on the basis of the literature cited. Personally, the analysis of comparisons of urban and suburban urban bird communities as well as the urbanization gradient lacks of references to a series of studies on the impact of ecological factors or urbanization gradient on the structure of avifauna (eg authors' teams such as Pennington', Blair', Emlen', Lepczyk' (North America), Jokimaki', Luniak', Tomiałojć', Witt' (Europe), in the context of additional food sources and their impact on wintering and breeding avifauna (eg Grubb' - North America, Brittingham', Cowie', Gaston', Fuller', Tryjanowski', Van Balen' - Europe), some works on long-term changes in avifauna of urban areas under the influence of environmental changes and progressive urbanization.

Validity of the findings

The work contains a new perspective on the research issues, potentially interesting.
The list of analyzed works in the context of the issues should be supplemented.

The conclusions should result from the analysis - this aspect depends on the additions and editing of the work.

Additional comments

A valuable analysis would be to focus not only on the analysis of species composition as such, but also to consider the impact of urban environment factors on the diversity of the ecological composition (food or nest guilds of birds communities).
The work in my opinion should have been published after a thorough rewording – precisely defined objectives allowing better definition of problems, a clear data presentation (similar as in Table 1), supplemented with a review of source publications, derivation of conclusions from analyzes.

Reviewer 3 ·

Basic reporting

no comment

Experimental design

no comment

Validity of the findings

no comment

Additional comments

I think that you are addressing very interesting and important topic with potentially important ecological consequences for birds. However, I think your manuscript does not follow the methodological assumptions most often used in case of reviews. First of all, it seems that your literature check has been performed without any clear methodology or you are just not providing info on that. You do not provide any details concerning the search (e.g. dates), and you just provide some examples of key words used in Google Scholar. This is not enough; you need to provide all the details concerning the search as well as present the results in detail: how many papers confirm a given process, etc. Did you summarize all the papers referring to your aims in the table 1? Anyway, this is not explained clearly in the ms. It is hard to believe that you found only 3 papers showing that artificial light affects nocturnal foraging in birds, as I found – during a very quick search – several others that are not included in your table (e.g. Byrkjedal, I., Lislevand, T. and Vogler, S. 2012. Do passerine birds utilise artificial light to prolong their diurnal activity during winter at northern latitudes? – Ornis Norvegica 35: 37–42. or Ockendon, N., Davis, S. E., Miyar, T. and Toms, M. P. 2009a. Urbanization and time of arrival of common birds at garden feeding stations. – Bird Study 56: 405–410 or Clewley, G. D., Plummer, K. E., Robinson, R. A., Simm, C. H. and Toms, M. P. 2015. The effect of artificial lighting on the arrival time of birds using garden feeding stations in winter: a missed opportunity? – Urban Ecosyst. doi: 10.1007%2Fs11252-015-0516-y). So why didn’t you include those papers as well? I suspect that your Google Scholar search is rather poor, perhaps because you included only 300 first papers (sorted by relevance or what?). I do not follow the logic behind the table 1 and papers selection you used – this should be described with much more details and in systematic manner. So this is my main comment to your study which I think should be addressed before the manuscript can be considered for publication. Below I also add some minor comments. But I still think your general idea was very interesting and thus I suggest improvement of the manuscript.

Introduction as a whole – to me more details are needed to put your study into context and justify better the need for the review. You need to present biological mechanisms potentially important for the temporal homogenization of the communities in response to stable resources. It would be also good to mention some local adaptations and provide some examples concerning birds foraging at night or wintering individuals in cities etc. As PeerJ is not an ornithological journal it would be also good to provide some justification why birds are good and important for such studies and how can we infer about other groups of species based on ornithological research.

28 – maybe add three „time” scales consider
42-44: I would delete these sentence, they do not add too much (further research are always needed etc.). Maybe just write directly which questions should be addressed in the future instead of writing that such questions are listed in the discussion.
51: double check but I would prefer ‘its’ instead of ‘his’
59-65: ok, but give examples and describe better the mechanisms behind the positive and negative response to resources stabilization. This issue is central for your study and justifies your work thus should be presented with more details.
76: I would delete the last aim, as this is only part of the interpretation so this is not a strictly scientific aim. Just write that you also give some recommendations for the future studies.
79: much more details concerning methodology is needed. Please provide dates when the search was conducted, write how did you analysed the results, how many articles in total did you consider, what was the distribution and range of publication year of these articles etc. Just saying that you searched google scholar is not enough – you need to provide detailed description of the methods used as your work could be potentially repeated by other researchers in the future.
91: don’t write about effects on humans – as far as I understand here you present your results so please focus on your result and skip other issues in this chapter.
126: well, not necessarily as you don’t have data on absolute amount of invertebrates, so this statement seems to be unjustified.
176: you write “most studies” and refer to table 1B but in the table all studies report negative effects so the question is: are all studies included in the table or is this just a subset of studies found during the search. This must be clarified.
191: did you mean Table 1C?
230-231: why do you think so? Could you provide more details or references supporting this statement?
241: if you consider invertebrates as a food for birds this chapter could be presented earlier, as a part of the chapter concerning habitat stability in cities. Now it is a bit strange to discuss stability of invertebrates and other taxa just after the discussion concerning birds.
250-256: I don’t understand why you provide data concerning mammals – it is rather hard to find a link between brown bears in garbage dump in Turkey to birds in cities. I would focus on the main aim of the study and delete all such inserts.

Figures – I don’t think your figures are useful. Fig 1 is very simple and shows only general conception of possible effects, with no clear link to your findings based on the literature review. In your analyses you don’t provide any clear justification nor evaluation of the effects presented on the figure so these are just your presumptions, not results supported by references. Fig 2 does not show any clear vegetation changes except some differences in colours of the photographs, which can also be driven by the weather conditions (the differences between two pictures taken in city are also visible). So this is not a proof for the stability-urbanization relationship. Fig3 – figure caption is unclear (what does “species that changed’ mean?), and showing example of species that were somehow related with urbanization does not bring too much to our understanding of the whole mechanism.

Table 1 is not very informative, and it is really hard to follow what you want to show here. Tables headings are lacking and it is hard to distinguish columns. Why horizontal line separates riparian sites from remaining part of the table? Not clear. Why study length is only given for the C section of the table – it seems that a substantial part of the data in the table 1 is missing? Why you present only a few papers out of 300 returns from your search? And many other similar questions.

---

## Round 0.2 · Minor Revisions

I have read the MS once again and answers to referee comment. Generally, this version is much better, but I still suggest some, relatively small changes. First of all, I rather suggest to change the title because in fact is not "a review" in classical sense, but more like an opinion paper. So in place "a review" I suggest to replace by something like this "a view from Latin America" and change also the introduction in that way. Seriously, is more necessary to promote urbanization vision from Southern Hemisphere that one more review on the process.
Also the second figure should be improved, and reduce use technical terms. Maybe small pictures in place of names? Like trees, buildings - to make the view more attractive to readers and then to promotion.

---

## Round 0.3 · accepted · Accept

Thank you for your revised submission. I am satisfied that you have addressed all the reviewer comments in your edits and rebuttal letter. I have expertise in this area and I did not feel that it was necessary to send it back to the reviewers for confirmation.

#